# Are Magnesium Screws Proper for Mandibular Condyle Head Osteosynthesis?

**DOI:** 10.3390/ma13112641

**Published:** 2020-06-10

**Authors:** Marcin Kozakiewicz

**Affiliations:** Department of Maxillofacial Surgery, Medical University of Lodz, 1st Gen. J. Haller Pl., 90-647 Lodz, Poland; marcin.kozakiewicz@umed.lodz.pl

**Keywords:** mandible, condylar head fracture, fixation material, screw, magnesium, titanium

## Abstract

Recently, magnesium alloys have gained a significant amount of recognition as potential biomaterials for degradable implants for craniofacial bone screws. Purpose: The aim of this work was to compare screws made specifically for mandibular head osteosynthesis from different materials. Materials and Methods: Screws measuring 14 mm made by one manufacturer specifically for mandibular head osteosynthesis out of the following materials were selected: magnesium (MgYREZr), titanium (Ti6Al7Nb), and polymer (PLGA). The axial pull-out strength and torsional properties were investigated. Results: Each type of screw presented different pull-out forces (Kruskal–Wallis test, *p* < 0.001). The magnesium screw had the highest pull-out force of 399 N (cracked without the screw out being pulled out), followed by the titanium screw, with a force of 340 N, and the PLGA screw, with a force of 138 N (always cracked at the base of the screw head without the screw being pulled out). ANOVA was performed for the maximal torques before damage to the screw (torsional properties), revealing that the maximal torque of the magnesium screw was 16 N·cm, while that of the titanium screw was 19 N·cm. The magnesium screw was significantly weaker than the titanium screw (*p* < 0.05). The measured torque and pull-out force were not related to each other (*p* > 0.05). Conclusion: Among the screws compared, the metal biodegradable magnesium screw seems to be the most suitable material for multiscrew mandibular head osteosynthesis, considering the condition of the fragile screwdriver socket.

## 1. Introduction

Recently, magnesium alloys have gained a significant amount of recognition as potential biomaterials for degradable implants for craniofacial bone screws. Since the first study of magnesium alloys was published, which included five magnesium osteosynthesis cases in the mandibular condyle head [1], a few metallic resorbable screws made specifically for this application have become available.

The current knowledge on biomaterials is being transformed by the development of corrosion-resistant resorbable metals. The role of biodegradable implants is to heal a specific trauma; support tissue regeneration; and, finally, disappear through degradation in the biological environment. In the last decade, magnesium alloys, as a new class of materials, have shown great potential to be used for maxillofacial surgery and have received much attention owing to their biodegradability [2,3,4], antiinflammatory properties [5,6], antibacterial properties [7,8,9], and osteogenesis inductivity [10,11,12]. Resorbable fixation allows one to avoid second surgery, which is, for example, very important in hemostasis-compromised patients in whom the next surgery is especially counter indicated [13].

Unfortunately, the mechanical properties of medical magnesium alloys are poorer than those of titanium. It is possible to damage screws during fixation (Figure 1). Because these incidents can occur, the author of this study has proposed modifications in screw design to reinforce screws and achieve the dimensions given in Table 1. However, it is still important to compare new metal resorbable screws with standardized titanium and long-established polymers. Therefore, the aim of this work was to compare screws made specifically for mandibular head fixation from different materials.

## 2. Materials and Methods

The tests included 35 screws (Table 1) made specifically for osteosynthesis of the mandible head by ChM (www.chm.eu; Juchnowiec Koscielny, Poland) from the following materials: magnesium alloy (14 pieces), titanium alloy (14 pieces), and polymer (7 pieces). The magnesium alloy utilized for manufacturing screws was MgYREZr (WE43MEO). The chemical composition of the magnesium screw by percentage of weight was 3.5–4.5% yttrium, 2.5–3.5% rare earth elements, 0.6% zirconium (max.), <200 ppm manganese, <200 ppm aluminum, <100 ppm silicon, <100 ppm copper, <80 ppm iron, <30 ppm nickel, and <20 ppm beryllium, and the remainder was magnesium. Titanium reference screws were made of Ti6Al7Nb alloy according to standard ISO 5832-11. The polymer screws used for the test were made of poly(L-lactide-glycolide), i.e., a copolymer of L-lactide and glycolide in a molar ratio of 85:15.

Solid polyurethane foam blocks were utilized in this study (Figure 2). The high variability in the density and elastic modulus of the bone affects the results of biomechanical tests [13]. Compared with cadaver bone, synthetic foam materials have been shown to yield less intra- and interspecimen variability (www.astm.org/Standards/F1839.htm). Foam blocks have consistent material properties that are similar to those of human bone. Solid polyurethane foam is widely used to mimic and is an ideal medium for mimicking human bone, and the American Society for Testing and Materials [14,15] has established it to be a standard material for testing orthopedic devices and instruments. In this study, polyurethane foam with a density of 0.64 g/cm^3^ (Sawbones Europe AB, Krossverksgatan, Malmö, Sweden) was used as a substitute for bone [16,17,18,19]. This material was chosen because of the cortico-cancellous structure of the mandible head.

The test methods used followed standard F543 (www.astm.org/Standards/F543.htm) for medical bone screws. This test method was used to compare the axial pull-out strength of three types of mandible head fixation screws (21 screws investigated) and to determine the torsional properties of metallic bone screws, i.e., magnesium versus titanium mandible head screws only (14 screws investigated).

An MTS Insight 100 kN testing system with an electromechanical drive system was used to determine the axial pull-out strength of medical bone screws: the force detected with the Interface 1010ACK-1.25KNB model was 1.25 kN, and the displacement detected was ±50 mm (MTS Insight 100, MTS Systems 14,000 Technology Drive, Eden Prairie, MN, USA). The application software utilized was TestWorks 4 (MTS Systems 14,000 Technology Drive, Eden Prairie, MN, USA). The test velocity used was 5 mm/min. at an ambient temperature of 23 ± 2 °C. MgYREZr, Ti6Al7Nb, and PLGA screws were tested at an insertion depth of 6 mm into a polyurethane block.

The MTS Bionix Servohydraulic test system (MTS Systems 14,000 Technology Drive, Eden Prairie, MN, USA) was used to determine the torsional properties of the metallic screws. The torsional configuration of the tabletop system enables the torsional moments to reach 150 Nm (detector MTS 662.20H-04) and the total rotation to reach ±140° (detector MTS ADT 605). The application software utilized was multipurpose (MTS Systems 14,000 Technology Drive, Eden Prairie, MN, USA). The applied velocity was 360°/min. at an ambient temperature of 23 ± 2 °C. MgYREZr and Ti6Al7Nb screws were tested.

Statistical analysis was performed in Statgraphics Centurion 18 (Statgraphics Technologies Inc. The Plains City, VA, USA). ANOVA and the Kruskal–Wallis test were used for the screw comparison. A p-value of less than 0.05 was considered statistically significant.

## 3. Results

The axial pull-out force [N] and maximal torque [N·cm] results are presented in Table 2 and Figure 3. The Kruskal–Wallis test statistic for axial pull-out force by the screw was 17.84. Each type of screw presented a different pull-out force (*p* < 0.001). The magnesium screw had the highest pull-out force value, which was approximately 400 N (Figure 4). In addition, the pull-out force of the titanium screw was 340 N, and that of the MgYREZr screw was 399 N, without the screw being pulled out of the test block. For the PLGA screws, the test always ended with a crack at the base of the screw head. However, during the study, no PLGA screws were pulled out from the test block. The ANOVA results for the maximal torques before damage to the screws (torsional properties) are presented in Figure 5 and Table 2. Fisher’s coefficient was 37.8, and the magnesium screw was significantly weaker than the titanium screw (*p* < 0.05). The measured torque and pull-out force were not related to each other (*p* > 0.05).

## 4. Discussion

Three types of screws can be chosen for mandible condyle head osteosynthesis: Low-profile titanium screws [20,21,22], polymer screws [23,24,25], and magnesium screws, which have recently become available [1,26]. Titanium is now the gold standard for open rigid internal fixation (ORIF) due to its rigidity, but it is not desirable to have screws inside the skeleton after healing is complete. Borys and coworkers suggested that there is increased free radical generation as well as increased inflammation and apoptosis in the tissue surrounding titanium mandibular fixations. Additionally, exposure to medical titanium alloy induces apoptosis, especially in the periosteum [27]. Titanium implants cause oxidative and nitrosative stress as well as disturbances in mitochondrial activity [28,29]. For these reasons, screws should be removed after it is confirmed that the bone has healed; however, in the region of the mandible head, second surgeries are not technically simple to perform and are associated with a risk of paralysis of the facial nerve. Polymer biodegradable screws solve this problem.

Early monomeric forms of biodegradable implants, namely, poly-L-lactide, have been shown to be associated with delayed degradation (>5 years), leading to occurrences of foreign-body reactions, local fistulas, osteolytic lesions, and peri-implant fluctuant swelling [30,31,32]. However, with the development of advanced copolymers, self-reinforcing materials, and increased control over degradation rates [33,34,35,36], biodegradable implants have shown promising results. However, the mechanical strength of polymer fixation remains low [37], which is why biodegradable screws are normally thicker than metallic screws [38]. Finally, the application polymer screws require bone fragment reduction; provisional stabilization; drilling; tapping; and, finally, screwing. There are too many challenging steps for mandibular head osteosynthesis. Ultrasound-activated resorbable pins provide a little assistance [23]. Therefore, resorbable metal osteosynthesis, which involves both the elimination of the implant after the healing process (as in polymers) and the mechanical strength of the metals, has been addressed and considered as the third option.

WE43 magnesium alloy has been used in the clinic in screws [1,26,39], and most magnesium alloys are used primarily as screws for internal fracture fixation [40]. For fixation material, fundamentally, the screw needs to remain stable in the bone after osteosynthesis, and it needs to be screwed in a way that does not damage the material. These screws make ORIF successful. For these reasons, the axial pull-out strength of medical bone screws and the torsional properties were selected as parameters to be tested.

Titanium small fragment screws 1.7–1.8 mm in diameter [41] have been shown to have sufficient compressibility [22,42] and strength [43] for mandible head osteosynthesis. As shown by the tests performed, the strength of the magnesium screw is significantly higher than that of the polymer screw. At this point, it should be noted that screws of different diameters were tested. This is an obvious methodological limitation, but the screws shared a common medical purpose and length. In brief, 14 mm screws from one manufacturer that were made specifically for the osteosynthesis of the mandible head were tested. Most likely, the screws were made to have different diameters due to the differences in the mechanical properties of PLGA, MgYREZr, and Ti6Al7Nb. If the screws had the same diameter, it is clear that the PLGA screw would be the weakest and the Ti6Al7Nb screw would be the strongest. In addition, a thin PLGA screw would be useless for osteosynthesis of the mandible head. Therefore, it would probably not be possible to discover that the axial pull-out force is the highest for MgYREZr and not for Ti4Al7Nb.

The pull-out force being higher for the magnesium screw than for the titanium screw results from the screw design (shaft diameter and thread depth) and not the material differences. It is known [22] that the torque is directly proportional to the screw diameter. Regardless, the polymer screw has the lowest axial pull-out force. Although the magnesium screw is thicker (2.2 mm), the torsional properties of the narrower titanium screw (1.7 mm) are better. Therefore, on the basis of this study, a titanium screw should be selected if a screw with high resilience is needed.

It is worth pointing out a few clinical issues. For MgYREZr alloy (both ChM and Syntellix manufacturers use the same alloy), the operator must handle the screw gently. The mechanical properties of this magnesium alloy are quite similar to those of bones. It is worth imagining how much effort is required to screw a screw made of cortical bone into such a bone. It is possible to damage the screwdriver socket or bend a 1.7 mm solid screw (without cannulation) designed with a slim shaft (Figure 1). Next, if normal construction lag screws made of magnesium are used, in the head-to-shaft transition, there is a risk of cracking due to reduced torsional strength and stress being concentrated at this location. Moreover, if low-profile magnesium screws are used, the resorption time will be shorter. This shorter time can be initially considered an advantage. In Neff staff, since the initiation of ORIF for head fractures in 1993, clinicians have routinely removed fixation material (1.8 mm small fragment screws since 2007) during a second-look procedure, which is mainly performed to prevent or mobilize screw-associated intra- and periarticular scar formation [41,44,45]. For more conservative or less experienced surgeons, it may be beneficial to leave osteosynthesis material for spontaneous resorption. Finally, a clinical issue is the choice of material used for fixation. Titanium screws are still available. In complicated comminute fractures, it is possible to use titanium screws in places distant from the joint surface, and when there is a risk of through-and-through insertion (especially through the joint surface), they can be combined with magnesium screws, which will resorb over time.

Maxillary artery laceration during maneuvers can occur with impacted bone segments or during careless proximal head fragment reduction with substandard lateral pterygoid muscle preparation. Drilling with too long a drill is another problem. It may be observed in mandible head fixation as operator perforates higher part of the neck trying to put a screw too low, below the mandible head. Known mechanical properties of the screws allow one to avoid iatrogenic damage and extension of the procedure. Moreover, in patients with increased risk of venous thromboembolism or haemorrhagic diathesis, shortening the time of the procedure is very important in order to avoid stroke/bleeding complications. This is the second reason for using proven screws and using an effective insertion technique [13,46].

Because the solubility of alloying elements in magnesium is limited and biocompatibility and biodegradation must be considered for the design of new screws, the ability to improve the mechanical properties is very restricted [39]. Magnesium alloys with favorable mechanical properties are expected to attract more attention when combined with new alloy designs, heat treatment, and plastic deformation techniques; furthermore, an approach that combines the strengthening benefits of nanocrystallinity with those of amorphization has been recently introduced to yield a dual-phase material that exhibits near-ideal strength in room temperature [39]; using this new process, the mechanical properties of magnesium screws can be improved. In summary, the advantages of magnesium screws are as follows: they yield stable and compressive osteosynthesis, there is no need to remove bone after healing, and they can be left for resorption if protruding hardware or temporal fixation is used prior to permanent screw insertion. The disadvantages are that hydrogen is produced during resorption and the screwdriver socket is weak. The screwdriver socket should be handled gently, even though the magnesium screw is large and measures up to 2.2 mm.

## 5. Conclusions

Among the screws compared, the metal biodegradable magnesium screw seems to be the most suitable material for multiscrew mandibular head osteosynthesis, considering the condition of the fragile screwdriver socket.

## Figures and Tables

**Figure 1 materials-13-02641-f001:**
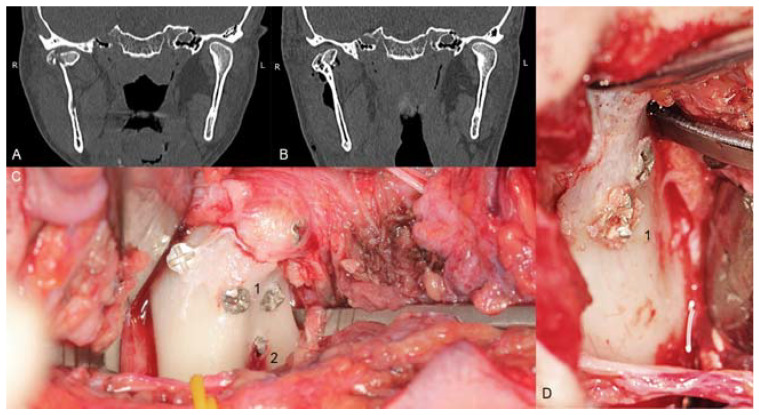
Examples of mechanical damage to the magnesium alloy screw. A and B—computed tomography images of the comminuted right head of the mandible (**A**) and images taken at the postoperative follow-up (**B**). Operative field of the same patients (**C**): screws inserted through the lateral pool and posterior head surface, and three screws inserted below the lateral pool. The *dexter caput mandibulae fractura* is shown in the next picture to the right (**D**): simple case of a Type B head fracture. Four of the 20 screws used became deformed. The decalibration of the slot for the screwdriver was observed (1) in all of them, once the core bend of 1.7 screws and the breaking of 1.7 screws was observed (2). Decalibration of the slot is not dangerous if the operator is aware of such circumstances, and these types of incidents can occur after the screw has been placed in the final position; however, bending and breaking of the screw compromises the integrity of osteosynthesis. All screws presented here are 1.7 mm system headless compressive screws (i.e., before the modification was tested in this study).

**Figure 2 materials-13-02641-f002:**
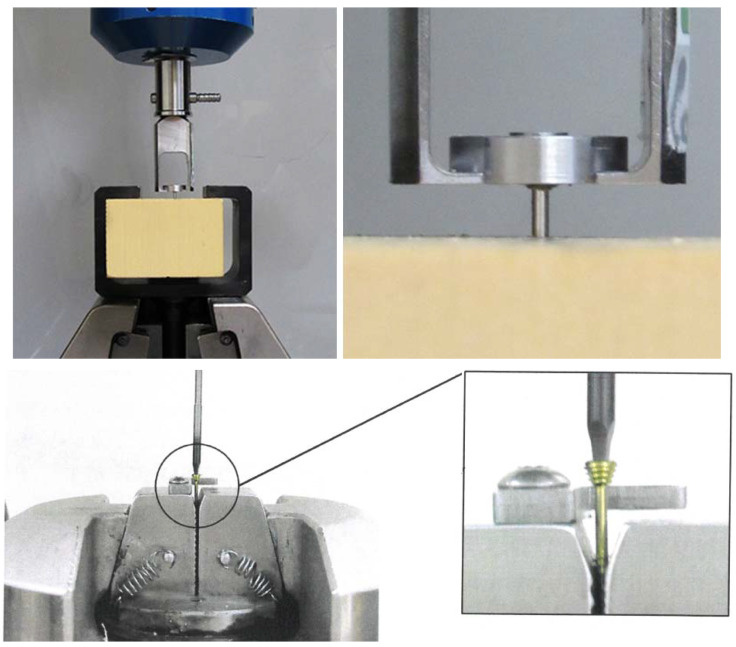
The MTS Insight 100 kN testing system with electromechanical drive system used for determining the axial pull-out strength (upper pictires) as well torsional properties (lower pictures) of medical bone screws.

**Figure 3 materials-13-02641-f003:**
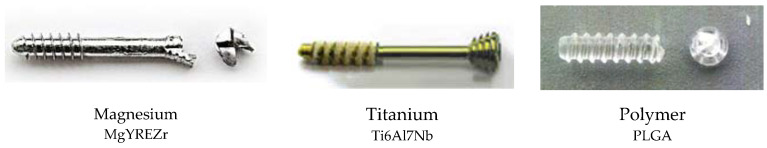
The screws after pull-out tests.

**Figure 4 materials-13-02641-f004:**
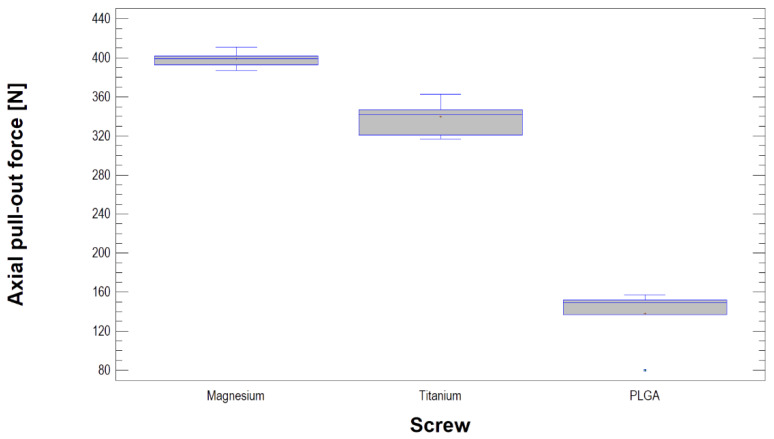
Summary statistics for axial pull-out force [N]. There were significant differences among the three types of screws (*p* < 0.05). Magnesium alloy screws show the greatest strength.

**Figure 5 materials-13-02641-f005:**
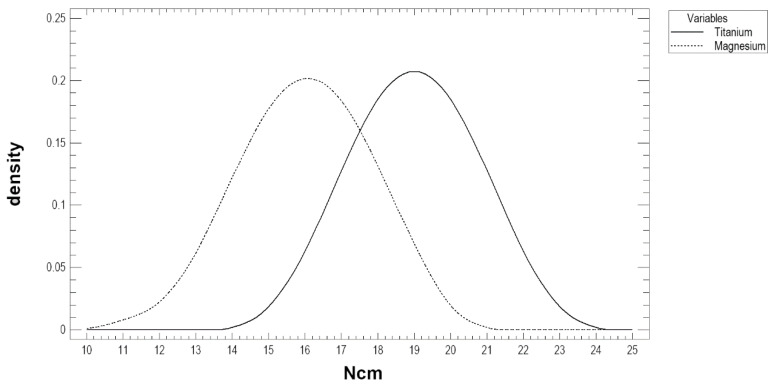
Maximal moment of the torque [N·cm] before the screw was damaged. The magnesium alloy screw is weaker than the titanium alloy screw (*p* < 0.05).

**Table 1 materials-13-02641-t001:** Comparison of 14 mm screws made by one manufacturer. All the screws shown below are used specifically for osteosynthesis of the mandible condylar head.

Screw Type	Thread Diameter	Core Diameter	Thread Pitch	Material	Appearance
Magnesium (Mg20 screw)	2.2 mm	1.5 mm	0.7 mm	MgYREZr	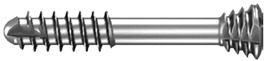
Titanium (W screw)	1.7 mm	1.1 mm	0.7 mm	Ti6Al7Nb	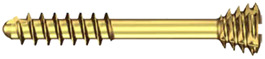
Polymer (Absorbable screw)	2.5 mm	1.9 mm	1.0 mm	PLGA	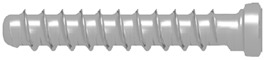

PLGA: poly(lactic-co-glycolic acid), Ti: titanium, Al: aluminum, Nb: niobium, Mg: magnesium, Y: yttrium, RE: rare earth elements, Zr: zirconium.

**Table 2 materials-13-02641-t002:** The axial pull-out strength of three types of mandible head osteosynthesis screws and the torsional properties of the metallic bone screws.

Parameter	Axial Pull-Out Force [N]			Maximal Torque [N·cm]	
	Polymer	Titanium	Magnesium	Titanium	Magnesium
Average ± standard deviation	138 ± 26.49	340 ± 15.92	399 ± 7.49	19 ± 0.82	16 ± 1.00
Minimum	80	317	387	18	14
Maximum	157	363	411	20	17
Range	77	46	24	2	3

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
