# Peer review of "Are Magnesium Screws Proper for Mandibular Condyle Head Osteosynthesis?"

_materials, 2020, doi:10.3390/ma13112641_

Round 1

Reviewer 1 Report

In this study, the author compares mechanically the screws made especially for the osteosynthesis of the mandibular head of magnesium, titanium and polymer. The subject is interesting, but for me it lacks a detailed comparison from a structural point of view, etc.

Author Response

Suggestions were made to change the ones given in the reviewer's table.

Reviewer 2 Report

The manuscript should be revised according to the following comments:

  1. In Table 1, besides the diameters of the used screws, please also indicate other dimensions (including the spacing of threads) of the screws.
  2. In line 81 and 82, the authors indicated form blocks have consistent materials are similar to human cancellous bone. Please give a table list for a comparison of physical properties between form and human cancellous bone (such as elastic modulus, hardness, tensile/compressive strength, density... etc.).
  3. In line 85, please give the unit of density in terms of g/cm3 instead of pcf.
  4. Mg alloys are chemical active and thought of biodegradable materials, and they also easily react with surrounding bone after implantation. In addition, Ti alloys are biocompatible materials, and a fairly chemical bonding can be formed between Ti alloys and hard tissues. However, there is no apparent chemical reaction or bonding between Mg, Ti specimens and the form in this study. Therefore, are the pull-out test and the pull-out forces valid for the application of MgYREZr and Ti6Al7Nd screws
  5. The authors did not show the figures of Mg, Ti and PLGA assemblies after pull-out tests.
  6. In Fig. 3, x-axis is usually for the "screw", and y-axis is usually for the pull-out force.
  7. What is th ephysical property or unit for the x-axis in Fig. 4?

Author Response

it is inluded in attached file

Reviewer 3 Report

The author had compared the pull out strength of magnesium , comparing it with titanium and polymer. The study is done well and has scientific soundness.Firstly, the variation in the diameter of the screws was my only concern. There will be variation in the pull out strength for different diameters.However, the author had clearly mentioned on the limitation of the different diameter in discussion section. Secondly , the bone block was mentioned as overall cancellous bone. It would had been still appropriate to use customised bone blocks using outer cortical and inner cancellous bone which would mimic the mandibular bone architecture. However, agreed to the fact that it is common to use these cancellous bone block. Lastly, Why was only 7 Polymer screws used ? whereas , 14 Ti and 14 Mg screws used ? Overall, a well written paper and interesting study useful in understanding the pull out strength of magnesium screws.

Author Response

it is included in attached file

Reviewer 4 Report

The manuscript deals with the very topical issue of magnesium screw. New biomaterials and their technical advantages are necessary in maxillofacial surgery. It is very important to research and compare specific properties. Methodical and statistical processing in the manuscript is correct. However, the discussions are missing some important parts related to post-operative complications in individual patients

In 4. Discussion, on page 7, lines 188-205. The authors should also discuss other postoperative complications and changes in hemostasis. It should describe bleeding complications after implantation. However, the manuscript is missing, describing the risks and management of patients with haemostasis disorders when implanting different types of screws. It is very difficult to manage patients with congenital bleeding disorders that require careful attention. It is necessary to emphasize the perioperative management of the patient with bleeding disorders. Cases of rare bleeding disorders such as congenital afibrinogenemia have been reported in the literature. The bleeding manifestations with gingival bleeding were repeated in this patient. An individual approach is needed if the patient needs a maxilofacial surgery. In 2016, a manuscript was published that related to a rare bleeding disorder and stressed that the surgery required thorough hematological management with the need for fine adjustment of balance between administration of replacement therapy and thromboprophylaxis. It is also appropriate to quote this publication Simurda T et al. Semin Thromb Hemost. 2016 Sep;42(6):689-92

The authors should also describe thromboprophylaxis in maxilofacial surgical interventions. Venous thromboembolism was the second most common medical complication, the second most common cause of increased length of hospital stay, the third most common cause of mortality and a significant increase in financial cost. Williams B et al. J Oral Maxillofac Surg. 2011 Mar;69(3):840-4.

Author Response

it is included in attached file

Round 2

Reviewer 1 Report

Accept in present form

Reviewer 4 Report

The presented manuscript has been corrected in response to the suggestions. The authors have followed the recommendations of the reviewers. After the revision, the provided data and interpretation of the results became more clear. I would like to thank the authors for resubmitting the manuscript and explaining the obscure points from the previous version. Now, the revised manuscript can be accepted for publication.